# Fast and stable determinant quantum Monte Carlo

**Carsten Bauer⋆**

Institute for Theoretical Physics, University of Cologne, 50937 Cologne, Germany

⋆ bauer@thp.uni-koeln.de

## Abstract

We assess numerical stabilization methods employed in fermion many-body quantum Monte Carlo simulations. In particular, we empirically compare various matrix decomposition and inversion schemes to gain control over numerical instabilities arising in the computation of equal-time and time-displaced Green's functions within the determinant quantum Monte Carlo (DQMC) framework. Based on this comparison, we identify a procedure based on pivoted QR decompositions which is both efficient and accurate to machine precision. The Julia programming language is used for the assessment and implementations of all discussed algorithms are provided in the open-source software library `StableDQMC.jl`.

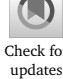

# 1 Introduction

Many-fermion systems play an important role in condensed matter physics. Due to their intrinsic correlations they feature rich phase diagrams which can not be captured by purely classical nor non-interacting theories. Especially at the lowest temperatures, quantum mechanical fluctuations driven by Heisenberg's uncertainty principle become relevant and lead to novel phases of matter like superconductivity and states beyond the Fermi liquid paradigm [1,2]. Because of the presence of interactions, predicting microscopic and thermodynamic properties of fermion many-body systems is inherently difficult. Analytical approaches are typically doomed to fail in cases where one can not rely on the smallness of an expansion parameter [3].

Fortunately, the determinant quantum Monte Carlo (DQMC) method [4–8] overcomes this limitation. The key feature of DQMC is that it is numerically exact - given sufficient computation time the systematical error is arbitrarily small. Provided the absence of the famous sign-problem [9,10], it allows for an efficient exploration of the relevant region of the exponentially large configuration space in polynomial time. It is an important unbiased technique for obtaining reliable insights into the physics of many-fermion systems which, among others, has been applied to the attractive and repulsive Hubbard model [11–13], the Kane-Mele-Hubbard [14], and metallic quantum criticality, including studies of antiferromagnetic [1–3,15], Ising-nematic [16], and deconfined quantum critical points [17] where fermionic matter fields are coupled to bosonic order parameters.

Although conceptually straightforward, care has to be taken in the implementation of DQMC because of inherent numerical instabilities arising from ill-conditioned matrix exponentials. Over time, stabilization schemes [5,8,18–20] based on various matrix factorizations, such as singular value decomposition (SVD), modified Gram-Schmidt, and QR decomposition, have been proposed for lifting these numerical issues. It is the purpose of this manuscript to review a subset of these techniques and to compare them with respect to accuracy and speed. Particular emphasis is placed on concreteness and reproducibility: we provide implementations of all discussed algorithms as well as the code to recreate all visualizations in this manuscript in form of the software library `StableDQMC.jl`. We choose the open-source, high-level programming language Julia [21,22] for our assessment which has proven [23,24] to be capable of reaching a performance comparable to established low-level languages in the field of numerical computing. Readers are invited to open issues and pull requests at the library

repository to discuss, improve, and extend the list of stabilization routines. Beyond reproducibility, the software library will also serve as an important abstraction layer allowing users to focus on physical simulation instead of numerical implementation details.

Specifically, the structure of the manuscript is as follows. We start by providing a brief introduction into the DQMC method in Sec. 2. In Sec. 3 we illustrate numerical instabilities in the DQMC and discuss their origin. Following this, we demonstrate (Sec. 4) how matrix factorizations can be utilized to remedy these numerical artifacts in chains of matrix products. In Sec. 5 we present and benchmark different schemes for stabilizing the computation of the equal-times Green's function, the fundamental building block in DQMC. Lastly, we turn to the calculation of time-displaced Green's functions in Sec. 6 before concluding and summarizing in Sec. 7.

## 2 Determinant Quantum Monte Carlo

We begin by reviewing the essentials of the determinant quantum Monte Carlo method [4–8]. We assume a generic quantum field theory that can be split into a purely bosonic part $S_B$ and a contribution $S_F$ from itinerant fermions. The latter comprises both fermion kinetics, $T$, and boson-fermion interactions, $V$. A famous example is given by the Hubbard model after a decoupling of the on-site interaction $U n_{i,\uparrow} n_{i,\downarrow}$ by means of a continuous Hubbard-Stratonovich or a discrete Hirsch transformation [25][1]. The quantum statistical partition function is given by

$$\mathcal{Z} = \int D\left(\psi, \psi^\dagger, \phi\right) e^{-S_B[\phi] - S_F[\psi, \psi^\dagger, \phi]}. \tag{1}$$

The first step in DQMC is to apply the quantum-classical mapping [26] and switch from the $d$ dimensional quantum theory above to a $D = d + 1$ dimensional classical theory. Here, the extra finite dimension of the classical theory is given by imaginary time $\tau$ and has an extent proportional to inverse temperature $\beta = 1/T$. Discretizing imaginary time into $M$ slices, $\beta = M\Delta\tau$, and applying a Trotter-Suzuki decomposition [27, 28] one obtains

$$\mathcal{Z} = \int D\phi \; e^{-S_B} \text{Tr}\left[\exp\left(-\Delta\tau \sum_{l=1}^{M} \psi^\dagger \left[T + V_\phi\right]\psi\right)\right]. \tag{2}$$

A separation of the matrix exponential then leads to a systematic error of the order $\mathcal{O}\left(\Delta\tau^2\right)$ in the partition function,

$$e^{A+B} \approx e^A e^B,$$

$$e^{-\Delta\tau(T+V)} \approx e^{-\frac{\Delta\tau}{2}T} e^{-\Delta\tau V} e^{-\frac{\Delta\tau}{2}T} + \mathcal{O}\left(\Delta\tau^3\right),$$

$$\mathcal{Z} = \int D\phi \; e^{-S_B} \text{Tr}\left[\prod_{l=1}^{m} B_l\right] + \mathcal{O}\left(\Delta\tau^2\right). \tag{3}$$

Here, $B_l = e^{-\frac{\Delta\tau}{2}\psi^\dagger T\psi} e^{-\Delta\tau \psi^\dagger V_\phi \psi} e^{-\frac{\Delta\tau}{2}\psi^\dagger T\psi}$ are imaginary time slice propagators. Note that the contribution $e^{-\Delta\tau \psi^\dagger V_\phi \psi}$ depends on the bosonic field $\phi$ due to a potential fermion-boson coupling in $V$. Rewriting the trace in (3) as a determinant [8] yields the fundamental form

$$\mathcal{Z} = \int D\phi \; e^{-S_B} \det G_\phi^{-1} + \mathcal{O}\left(\Delta\tau^2\right), \tag{4}$$

---

[1]Depending on the decomposition channel, the bosonic field $\phi$ represents either spin or charge fluctuations in this case.

where

$$G = [\mathbb{1} + B_M B_{M-1} \cdots B_1]^{-1} \tag{5}$$

is the equal-time Green's function [26]. Accordingly, the Metropolis probability weight is given by

$$p = \min\left\{1, e^{-\Delta S_\phi} \frac{\det G}{\det G'}\right\}. \tag{6}$$

This implies that, considering a generic, global update one needs to compute the Green's function $G$ and it's determinant in each DQMC step[2].

Importantly, it is only under specific circumstances, such as the presence of a symmetry, that the integral kernel of the partition function can be safely interpreted as a probability weight since $G_\phi$ and its determinant are generally complex valued. This is the famous sign problem [29].

## 3 Numerical instabilities

To showcase the typical numerical instabilities arising in the DQMC framework we consider the Hubbard model in one dimension at half filling,

$$H = -t \sum_{\langle i,j \rangle} c_i^\dagger c_j + U \sum_i \left(n_{i\uparrow} - \frac{1}{2}\right)\left(n_{i\downarrow} - \frac{1}{2}\right), \tag{7}$$

which is free of the sign-problem [29]. We set the hopping amplitude to unity, $t = 1$[3].

As seen from Eq. (5), the building block of the equal-time Green's function is a matrix chain multiplication of imaginary time slice matrices. To simplify our purely numerical analysis below we assume that these slice matrices $B_i$ are independent of imaginary time,

$$B(\beta, 0) \equiv B_M B_{M-1} \cdots B_1 = \underbrace{BB \cdots B}_{M \text{ factors}}, \tag{8}$$

which, physically, amounts to assuming a constant bosonic field $\phi = const$.

First, we consider the non-interacting system, $U = 0$. As apparent from Fig. 1, a naive computation of Eq. 8 fails for $\beta \geq \beta_c \approx 10$. Leaving a discussion of the stabilization of the computation for the next section, let us highlight the origin of this instability. The eigenvalues of the non-interacting system are readily given by

$$\epsilon_k = -2t \cos(k), \tag{9}$$

such that energy values are bounded by $-2t \leq \epsilon_k \leq 2t$. A single positive definite slice matrix $B = e^{-\Delta \tau T}$ therefore has a condition number of the order of $\kappa \approx e^{4|t|\Delta \tau}$ and, consequently, $B(\tau, 0)$ has $\kappa \approx e^{4|t|M\Delta \tau} = e^{4|t|\beta}$. This implies that the scales present in $B(\tau, 0)$ broaden exponentially at low temperatures $T = 1/\beta$ leading to inevitable roundoff errors due to finite machine precision which spoil the result.

We can estimate the expected inverse temperature of this breakdown for the data type `Float64`, that is double floating-point precision according to the IEEE 754 standard [30], by solving $\kappa(\beta) \sim 10^{-17}$ for $\beta_c$. One finds $\beta_c \approx 10$ in good agreement with what is observed in Fig. 1a. Switching to the data type `Float128`[4] (quadruple precision) with $\beta_c \approx 20$ in Fig. 1b, the onset of roundoff errors is shifted to lower temperatures in accordance with expectations.

---

[2]For local updates one can typically avoid those explicit calculations and compute the ratio of determinants in Eq. (6) directly [2].

[3]We will consider the canonical discrete decoupling [25] in the spin channel due to Hirsch in our analysis.

[4]The datatype `Float128` is provided by the Julia package `Quadmath.jl`.

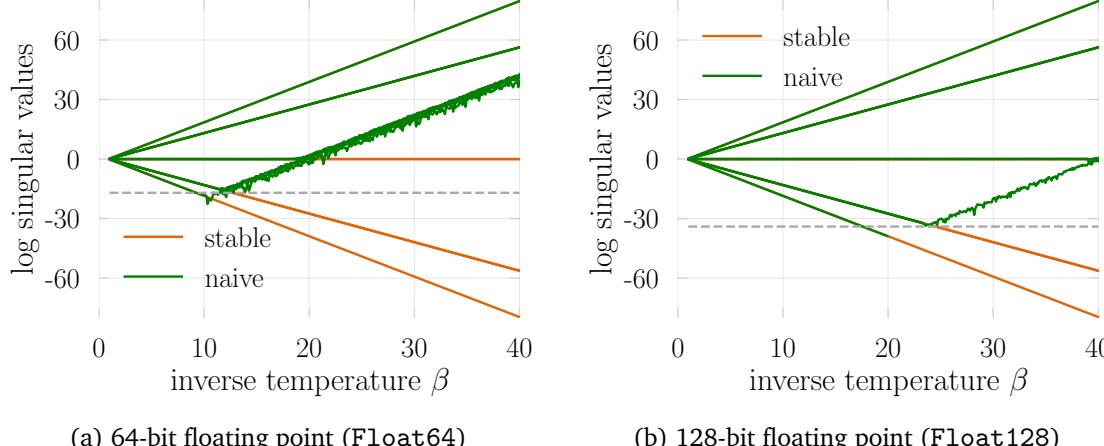

(a) 64-bit floating point (`Float64`)     (b) 128-bit floating point (`Float128`)

Figure (1) **Numerical instabilities** due to finite machine precision arising in the calculation of the time slice matrix chain product $B_M B_{M-1} \cdots B_1$ for model (7). Different lines represent logarithmic singular values as observed in naive (green) and arbitrary precision computations (orange) for a $N = 4$ system. Due to (quasi) degeneracies only 5 out of 8 singular values are visually distinguishable. The dashed line (grey) indicates the expected floating point accuracy[5].

## 4 Stabilization: time slice matrix multiplications

### 4.1 Stabilization scheme

A trivial solution to the issue outlined above is to perform all numerical operations with arbitrary precision. In Julia, this can be realized by means of the `BigFloat` data type[6]. However, this comes at the expense of (unpractical) slow performance due to algorithmic overhead and lack of hardware support. Arbitrary precision numerics is nevertheless a valuable tool and we will use it to benchmark the accuracy of stabilization methods below[7].

How can we get a handle on the numerical instabilities in a floating point precision computation? As has been realized [18] soon after the introduction of the DQMC method [4], an effective strategy is to keep the broadly different scales in the matrix exponentials separated throughout the computation (as much as possible) and only mix them in a final step, if necessary. A useful tool for extracting the scale information is a matrix decomposition,

$$B = UDX. \tag{10}$$

Here, $U$ and $X$ are matrices of order unity and $D$ is a real diagonal matrix hosting the exponentially spread scales of $B$. We will refer to the values in $D$ as singular values independent of the particular decomposition. Using Eq. (10), we can stabilize the matrix multiplication of slice matrices $B_1$ and $B_2$ in Eq. (8) as follows [18], (`fact_mult` in `StableDQMC.jl`)

$$B_2 B_1 = \underbrace{U_2 D_2 X_2}_{B_2} \underbrace{U_1 D_1 X_1}_{B_1} = U_2 \underbrace{(D_2((X_2 U_1)D_1))}_{U'D'X'} X_1) = U_r D_r X_r. \tag{11}$$

---

[5]We estimate the precision as $p = \log_{10}(2^{\text{fraction}})$, where fraction is the mantissa of a given binary floating point format. This gives $p \sim 16$ for `Float64` and $p \sim 34$ for `Float128`.

[6]Technically, `BigFloat` has a finite, arbitrarily high precision.

[7]For our non-interacting model system one can alternatively simply diagonalize the Hamiltonian and calculate the Green's function exactly.

Here, $U_r = U_2 U'$, $D_r = D'$, $X_r = X' X_1$, and $U'D'X'$ indicates an intermediate matrix decomposition. If we follow this scheme, in which parentheses indicate the order of operations, largely different scales present in the diagonal matrices won't be additively mixed throughout the computation. Specifically, note that the multiplication of the well-conditioned, combined, unit-scale matrix $U = X_2 U_1$ with $D_1$ and $D_2$ does preserve the scale information: the diagonal matrices merely rescale the columns and rows of $U$,

$$D_2 U D_1 = \begin{bmatrix} \mathbf{S} & & & \\ & \mathbf{S} & & \\ & & \mathsf{s} & \\ & & & {\scriptstyle \mathsf{s}} \end{bmatrix} \underbrace{\begin{bmatrix} \mathsf{s} & \mathsf{s} & \mathsf{s} & \mathsf{s} \\ \mathsf{s} & \mathsf{s} & \mathsf{s} & \mathsf{s} \\ \mathsf{s} & \mathsf{s} & \mathsf{s} & \mathsf{s} \\ \mathsf{s} & \mathsf{s} & \mathsf{s} & \mathsf{s} \end{bmatrix}}_{\text{unit scale}} \begin{bmatrix} \mathbf{S} & & & \\ & \mathbf{S} & & \\ & & \mathsf{s} & \\ & & & {\scriptstyle \mathsf{s}} \end{bmatrix} \tag{12}$$

$$= \begin{bmatrix} \mathbf{S} & & & \\ & \mathbf{S} & & \\ & & \mathsf{s} & \\ & & & {\scriptstyle \mathsf{s}} \end{bmatrix} \begin{bmatrix} \mathsf{s}\mathbf{S} & \mathsf{s}\mathbf{S} & \mathsf{s}^2 & {\scriptstyle \mathsf{s}\mathsf{s}} \\ \mathsf{s}\mathbf{S} & \mathsf{s}\mathbf{S} & \mathsf{s}^2 & {\scriptstyle \mathsf{s}\mathsf{s}} \\ \mathsf{s}\mathbf{S} & \mathsf{s}\mathbf{S} & \mathsf{s}^2 & {\scriptstyle \mathsf{s}\mathsf{s}} \\ \mathsf{s}\mathbf{S} & \mathsf{s}\mathbf{S} & \mathsf{s}^2 & {\scriptstyle \mathsf{s}\mathsf{s}} \end{bmatrix} \tag{13}$$

$$= \begin{bmatrix} \mathbf{S}^2{\scriptstyle \mathsf{s}} & \mathbf{S}\mathbf{S}{\scriptstyle \mathsf{s}} & \mathbf{S}\mathsf{s}^2 & \mathbf{S}{\scriptstyle \mathsf{s}\mathsf{s}} \\ \mathbf{S}\mathbf{S}{\scriptstyle \mathsf{s}} & \mathbf{S}^2{\scriptstyle \mathsf{s}} & \mathbf{S}\mathsf{s}^2 & \mathbf{S}{\scriptstyle \mathsf{s}\mathsf{s}} \\ \mathbf{S}\mathsf{s}^2 & \mathbf{S}\mathsf{s}^2 & \mathsf{s}^3 & \mathsf{s}^2{\scriptstyle \mathsf{s}} \\ \mathbf{S}{\scriptstyle \mathsf{s}\mathsf{s}} & \mathbf{S}{\scriptstyle \mathsf{s}\mathsf{s}} & \mathsf{s}^2{\scriptstyle \mathsf{s}} & {\scriptstyle \mathsf{s}\mathsf{s}}^2 \end{bmatrix} . \tag{14}$$

Repeating the procedure (11), we obtain a numerically accurate $UDX$ decomposition of the full time slice matrix chain $B(\tau, 0)$, which preserves the scale information as indicated in Fig. 1.[8] We note in passing that in practice it is often unnecessary to stabilize every individual matrix product. Instead one typically performs a mixture of naive and stabilized products for the sake of speed while still retaining numerical accuracy [8].

## 4.2 Matrix decompositions

There are a various matrix decompositions that one could employ to obtain the factorization $B = UDX$, Eq (10). In the following we will consider the two most popular choices in DQMC codes [7, 8, 18].

### 4.2.1 SVD ($UDV^\dagger$)

The singular value decomposition (SVD) is given by

$$B = USV^\dagger, \tag{15}$$

where $U$ and $V^\dagger$ are unitary and $S$ is real and diagonal.

For computing the SVD of a matrix of regular floating point precision (`Matrix{Float64}`), Julia utilizes the heavily optimized routines provided by LAPACK[9] [31]. Concretely, there exist three different implementations of SVD algorithms [32]:[10]

- `gesdd` (default): Divide-and-conquer (D&C)

---

[8]Note that we do not discuss the faster way to calculate $B^M$ as $UD^M X$. This is intentional since most real systems will involve fermion-boson interactions and the slice matrices will depend on $\phi(\tau)$.

[9]We will report on results obtained with the LAPACK implementation OpenBLAS that ships with Julia. Qualitatively similar results have been found in an independent test based on Intel's Math Kernel Library (MKL).

[10]Note that the names of LAPACK functions typically encode properties of the input matrix such as realness or symmetry. In Julia multiple-dispatch takes care of routing different matrix types to different *methods*. The Julia function `gesdd` works for both real and complex matrices, that is there is no (need for) `cgesdd`.

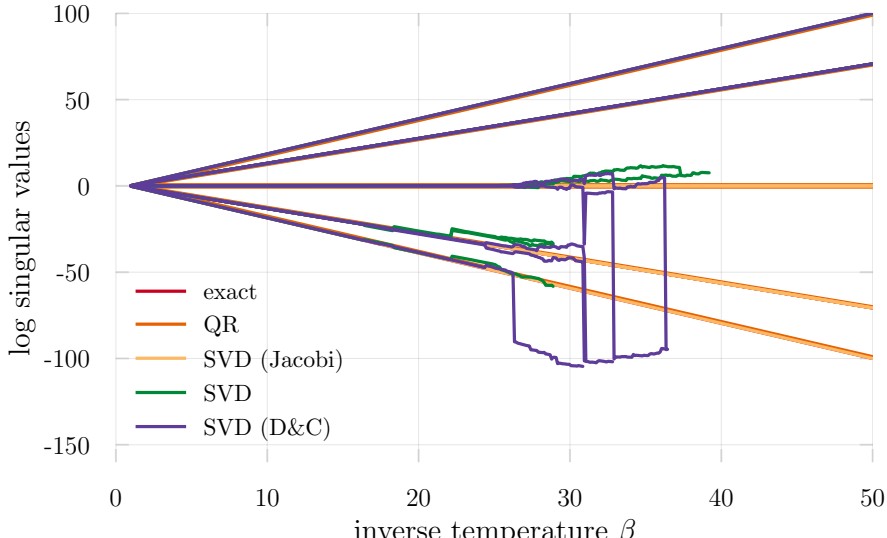

Figure (2) **Comparison of matrix decompositions** to heal the numerical instabilities in the calculation of the time slice matrix chain product $B_M B_{M-1} \cdots B_1$ for model (7). Different lines represent logarithmic singular values as observed in stabilized computations. The QR (orange) and Jacobi SVD singular values (yellow) lie on top of the exact result (red) whereas both the regular SVD (green) and the divide-and-conquer SVD (purple) show large deviations at low temperatures $\beta \gtrsim 25$ ($\Delta\tau = 0.1$).

- gesvd: Bidiagonal QR iteration (conventional)

- gesvj: Jacobi algorithm (through JacobiSVD.jl)

To simplify the manual access to these algorithms we export convenience wrappers of the same name in StableDQMC.jl. We will compare all three variants below and benchmark them against an arbitrary precision computation using BigFloat. Since LAPACK doesn't support special number types, we will utilize the native-Julia SVD implementation provided by GenericSVD.jl in this case.

### 4.2.2 QR ($UDT$)

A QR decomposition reads

$$B = QR = UDT, \tag{16}$$

where $Q$ is unitary, $R$ is upper triangular, and we have split $R$ into a diagonal part $D$ and an upper triangular part $T$ in the second step. Specifically, $U = Q$ is unitary, $D = \text{diag}(R)$ is real and diagonal, and $T$ is upper triangular.

In Julia, one can obtain the $QR$ factored form of a matrix using qr from the standard library LinearAlgebra. We will consider the pivoted QR, which is deployed in the public DQMC implementations ALF [33] and QUEST [34], in form of LAPACK's geqp3 in our analysis. A factorization into $UDT$ form is provided by functions udt and udt! in StableDQMC.jl.

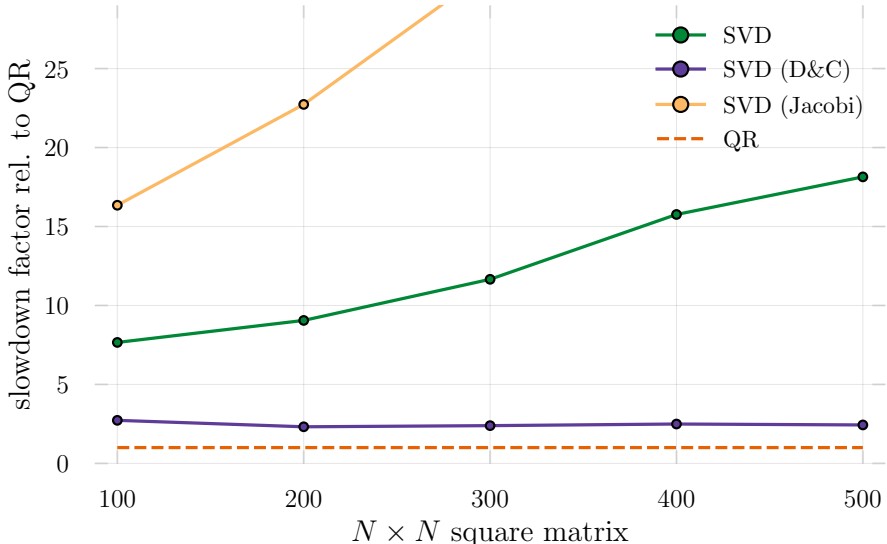

Figure (3) **Computational efficiency of matrix decompositions**. Shown is the runtime cost of the factorization of a complex matrix of size $N \times N$ by means of various SVD algorithms relative to the QR decomposition.

### 4.3 Benchmarks

#### 4.3.1 Accuracy

Supplementing our general considerations above, we test the correctness of the matrix product stabilization procedure with respect to varying the concrete SVD and QR factorization algorithms. Fig. 2 shows the logarithmic singular values of the time slice matrix chain $B(\beta, 0)$ as a function of inverse temperature $\beta$ obtained from employing different matrix decompositions. Clearly, the accuracy of the computed singular values shows a strong dependence on the chosen factorization algorithm. While the results for the QR decomposition and Jacobi SVD seem to fall on top of the exact result, we observe large deviations for the conventional and D&C SVD algorithms. This effect is particularly pronounced at low temperatues, $\beta \gtrsim 25$. The fact that small scales are lost in these SVD variants, while large ones are still correct, can be understood from LAPACK's SVD error bounds [35]: The error is bounded relative to the largest singular value. Thus, large scales are computed to high relative accuracy and small ones may not be.

#### 4.3.2 Efficiency

Turning to computational efficiency, we illustrate runtime cost measurements for all considered SVD variants relative to the QR decomposition in Fig. 3. We find that both the conventional SVD and Jacobi SVD are an order of magnitude slower than the QR decomposition while only the divide-and-conquer algorithm shows comparable speed. Among the SVD variants, the Jacobi SVD is the most costly by a large margin, having about twice the runtime of the conventional SVD for small system sizes.

## 5 Stabilization: equal-time Green's function

Similar to the considerations above, a naive computation of the Green's function according to Eq. (5) is potentially unstable because of numerical roundoff errors due to finite machine

precision. In particular, adding the identity to the ill-conditioned slice matrix chain $B(\tau,0)$ will generally wash out small singular values and will lead to a non-invertible result such that the subsequent inversion in Eq. (5) is ill-defined. This clearly prohibits a safe calculation of the equal-time Green's function and asks for numerical stabilization techniques.

## 5.1 Inversion schemes

As for the time slice matrix products in Eq. (11), the strategy will be to keep exponentially spread scales as separated as possible. A straightforward scheme [7,8] (`inv_one_plus`) to add the unit matrix and perform the inversion of $\mathbb{1} + B(\tau,0)$ in a stabilized manner is given by

$$G = [\mathbb{1} + UDX]^{-1} = [U\underbrace{(U^\dagger X^{-1} + D)}_{udx}X]^{-1} = [(Uu)d(xX)]^{-1} = U_r D_r X_r, \qquad (17)$$

where $U_r = (xX)^{-1}$, $D_r = d^{-1}$, and $X_r = (Uu)^{-1}$. Here, the intermediate addition (parentheses in the second line of (17)) of unit scales and singular values is separated from the unitary rotations such that $U^\dagger X^{-1}$ only acts as a clean cutoff,

$$U^\dagger X^{-1} + D = \begin{bmatrix} s & s & s & s \\ s & s & s & s \\ s & s & s & s \\ s & s & s & s \end{bmatrix} + \begin{bmatrix} \mathsf{S} & & & \\ & \mathsf{S} & & \\ & & s & \\ & & & \scriptstyle s \end{bmatrix} = \begin{bmatrix} \mathsf{S} & s & s & s \\ s & \mathsf{S} & s & s \\ s & s & s & s \\ s & s & s & s \end{bmatrix}. \qquad (18)$$

As we will demonstrate for the time-displaced Green's function in Sec. 6, a procedure like Eq. (17) based on a single intermediate decomposition will still fail to give accurate results for some of the matrix decompositions. For this reason, we consider another stabilization procedure put forward by Loh *et al.* [5,18] (`inv_one_plus_loh`), in which one initially separates the scales of the diagonal matrix $D$ into two factors $D_p = \max(D,1)$ and $D_m = \min(D,1)$,

$$D_p = \begin{bmatrix} \mathsf{S} & & & \\ & \mathsf{S} & & \\ & & s & \\ & & & s \end{bmatrix}, \quad D_m = \begin{bmatrix} s & & & \\ & s & & \\ & & s & \\ & & & \scriptstyle s \end{bmatrix}, \qquad (19)$$

and performs two intermediate decompositions,

$$G = [\mathbb{1} + UDX]^{-1} = [\mathbb{1} + UD_m D_p X]^{-1} = [(X^{-1}D_p^{-1} + UD_m)D_p X]^{-1} \qquad (20)$$

$$= X^{-1}[D_p^{-1}\underbrace{(X^{-1}D_p^{-1} + UD_m)^{-1}}_{udx}] = U_r D_r X_r, \qquad (21)$$

$$\underbrace{\phantom{= X^{-1}[D_p^{-1}(X^{-1}D_p^{-1} + UD_m)^{-1}]}}_{udx}$$

where $U_r = X^{-1}u$, $D_r = d$, and $X_r = x$.

## 5.2 Benchmarks

We assess how different matrix decomposition algorithms perform in stabilized computations of $B(\beta,0)$, the Green's function $G$ both with respect to accuracy and speed. All results are for the Hubbard model, Eq. 7, with the interaction strength set to $U = 0$ and $U = 1$ (alpha transparent in all plots)

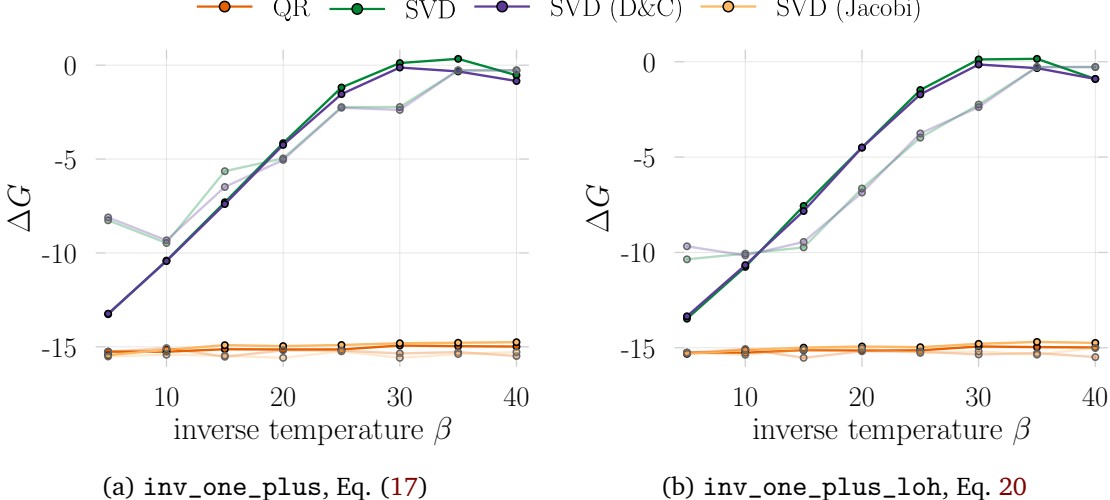

(a) `inv_one_plus`, Eq. (17)  (b) `inv_one_plus_loh`, Eq. 20

Figure (4) **Accuracy of the Green's function** obtained from stabilized compu-
tations using the listed matrix decompositions and inversion schemes. Shown is
$\Delta G = \log(\max(\mathrm{abs}(G - G_{\mathrm{exact}})))$ for $U = 0$ (solid) and $U = 1$ (alpha transparent).

### 5.2.1 Accuracy

Starting from a stabilized computation of $B(\beta, 0)$, Sec. 4, we calculate the equal-time Green's
function by performing the inversion according to the schemes outlined above and varying
the applied matrix factorization. In Fig. 4a we show our findings for `inv_one_plus`, Eq. 17,
where we have taken the maximum absolute difference between the computed and the exact
Green's function as an accuracy measure. At high temperatures and for $U = 0$, we observe that
all decompositions lead to a good approximation of $G_{\mathrm{exact}}$ with an accuracy close to floating
point precision. However, when turning to lower temperatures the situations changes dra-
matically. We find that only the QR decomposition and the Jacobi SVD deliver the Green's
function reliably. Compared to the other SVD variants, which fall behind by a large margin
and fail to reproduce the exact result, they consistently show about optimal accuracy even
in the presence of interactions. As displayed in Fig. 4b, switching to the inversion scheme
`inv_one_plus_loh`, Eq. 20, does generally improve the accuracy but deviations of the regu-
lar SVD and D&C SVD remain of the order of unity at the lowest temperatures.

These findings suggest that only the QR decomposition and the Jacobi SVD, irrespective of
the inversion procedure, are suited for computing the equal time Green's function in DQMC
reliably.

### 5.2.2 Efficiency

Independent of the employed inversion scheme, matrix decompositions are expected to be the
performance bottleneck in the Green's function computation. We hence expect the speed dif-
ferences apparent in Fig. 3 to dominate benchmarks of the full Green's function calculation as
well. This anticipation is qualitatively confirmed in Fig. 5, which shows the runtime cost of
the Green's function computation for both inversion schemes and all matrix decompositions
relative to the QR. While the divide-and-conquer SVD is in the same ballpark as the QR de-
composition the other SVD algorithms fall behind by a large margin (an order of magnitude)
for both inversion procedures. Importantly, this apparent runtime difference is increasing
with system size. The observation that the relative slowdown factor is larger for the inversion
scheme `inv_one_plus_loh` can be understood from the fact that it requires one additional
intermediate matrix decomposition.

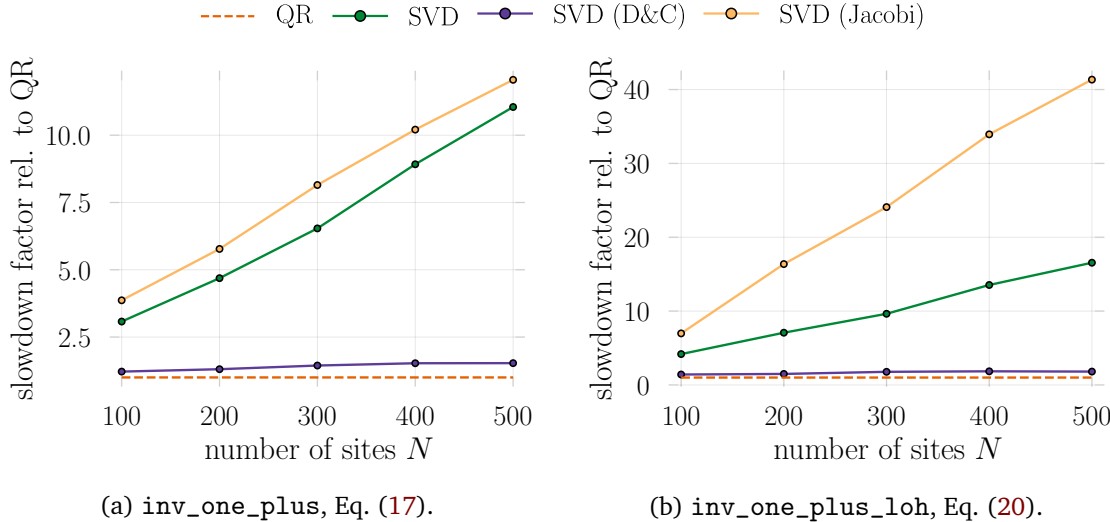

(a) `inv_one_plus`, Eq. (17).  (b) `inv_one_plus_loh`, Eq. (20).

Figure (5) **Efficiency of the stabilized Green's function calculation** using the listed matrix decompositions and inversion schemes. Shown are results for $U = 0$.

Combined with the accuracy results these findings suggest that among the QR decomposition and the Jacobi SVD, which are found to be reliable in both inversion schemes, the QR decomposition has a significantly lower runtime cost and is therefore to be preferred in DQMC.

## 6 Stabilization: time-displaced Green's function

In this section, we turn to the stabilization of time-displaced Green's functions. While these are not required in the basic DQMC, that is for generating a representative Markov chain of configurations, they are central to the measurement of time-displaced correlation functions such as pairing correlations and the superfluid density [6,7].

First, we generalize our definition of the Green's function, Eq. 5, to include imaginary time $\tau = l\Delta\tau$,

$$G(\tau) = \langle c_i c_j^\dagger \rangle_{\phi_l} = \left[\mathbb{1} + B_{l-1}\ldots B_1 B_M \ldots B_l\right]^{-1}. \tag{22}$$

Note that $G \equiv G_1 = G_{M+1} = \left[\mathbb{1} + B_M \ldots B_l\right]^{-1}$ due to fermionic boundary conditions. The time displaced Green's function can now be defined in terms of the time ordering operator $T$ as [7,8]

$$G_{l_1, l_2} \equiv G(\tau_1, \tau_2) \equiv \langle T c_i(\tau_1) c_j^\dagger(\tau_2) \rangle_\varphi.$$

More explicitly, this reads

$$G(\tau_1, \tau_2) = \begin{cases} B_{l_1} \cdots B_{l_2+1} G_{l_2+1}, & \tau_1 > \tau_2, \\ -\left(1 - G_{l_1+1}\right)\left(B_{l_2} \cdots B_{l_1+1}\right)^{-1}, & \tau_2 > \tau_1. \end{cases} \tag{23}$$

In principle, this gives us a prescription for how to calculate $G(\tau_1, \tau_2)$ from the equal time Green's function discussed in Sec. 5. However, when $|\tau_1 - \tau_2|$ is large a naive calculation of the matrix products in Eq. 23 would be numerically unstable, as seen in Sec. 4. More importantly, by first calculating the equal-time Green's function one already mixes (and looses) scale information in the last recombination step, $G = UDX$. It is therefore advantageous to

compute the time-displaced Green's function directly as (assuming $\tau_1 > \tau_2$ for simplicity)

$$G(\tau_1, \tau_2) = B_{l_1} \cdots B_{l_2+1} G_{l_2+1} \tag{24}$$

$$= B_{l_1} \cdots B_{l_2+1} \left[ \mathbb{1} + B_{l_2} \ldots B_1 B_M \ldots B_{l2+1} \right]^{-1} \tag{25}$$

$$= \left[ \underbrace{B_{l_2+1}^{-1} \cdots B_{l_1}^{-1}}_{U_L D_L X_L} + \underbrace{B_{l_2} \ldots B_1 B_M \ldots B_{l1+1}}_{U_R D_R X_R} \right]^{-1} \tag{26}$$

$$= \left[ U_L D_L X_L + U_R D_R X_R \right]^{-1}. \tag{27}$$

## 6.1 Inversion schemes

As for the equal time Green's function (Sec. 5), one must be careful to keep the scales in $D_L$ and $D_R$ separated when performing the summation and inversion to avoid unnecessary floating point roundoff errors. As a first explicit procedure, we consider a simple generalization of Eq. 17 (inv_sum),

$$
\begin{aligned}
G(\tau_1, \tau_2) &= [U_L D_L X_L + U_R D_R X_R]^{-1} \\
&= [U_L \underbrace{(D_L X_L X_R^{-1} + U_L^\dagger U_R D_R)}_{udx} X_R]^{-1} \\
&= [(U_L u) d^{-1} (x X_R)]^{-1} \\
&= U_r D_r X_r,
\end{aligned} \tag{28}
$$

where $U_r = (x X_R)^{-1}$, $D_r = d^{-1}$, and $X_r = (U_L u)^{-1}$.

Analogously, we can generalize the scheme by Loh *et al.* [5], Eq. 20, in which we split the scales into matrix factors $D_m = \min(D, 1)$, $D_p = \max(D, 1)$, (inv_sum_loh)

$$
\begin{aligned}
G(\tau_1, \tau_2) &= [U_L D_L X_L + U_R D_R X_R]^{-1} \\
&= [U_L D_{Lm} D_{Lp} X_L + U_R D_{Rm} D_{Rp} X_R]^{-1} \\
&= \left[ U_L D_{Lp} \underbrace{\left( \frac{D_{Lm}}{D_{Rp}} X_L X_R^{-1} + U_L^\dagger U_R \frac{D_{Rm}}{D_{Lp}} \right)}_{udx} X_R D_{Rp} \right]^{-1} \\
&= X_R^{-1} \frac{1}{D_{Rp}} \underbrace{[udx]^{-1} \frac{1}{D_{Lp}} U_L^\dagger}_{udx} \\
&= U_r D_r X_r,
\end{aligned} \tag{29}
$$

where $U_r = X_R^{-1} u$, $D_r = d$, and $X_r = x U_L^\dagger$.

We note that Hirsch [7, 36] has proposed an alternative method for computing the time-displaced Green's function based on a space-time matrix formulation of the problem. Although this technique has been successfully deployed in many-fermion simulations we won't discuss it here because of its subpar computational scaling: for a system composed of $N$ lattice sites, fermion flavors $f$, and imaginary time extent $M$ one has to invert (naively a $\mathcal{O}(x^3)$ operation) a matrix which takes up $\mathcal{O}((NMf)^2)$ memory. Similarly, Assaad *et al.* [8] have described an approach to compute both equal time and time-displaced Green's functions in one step. However, this requires to work with extended matrices of doubled linear dimension compared to the regular Green's functions.

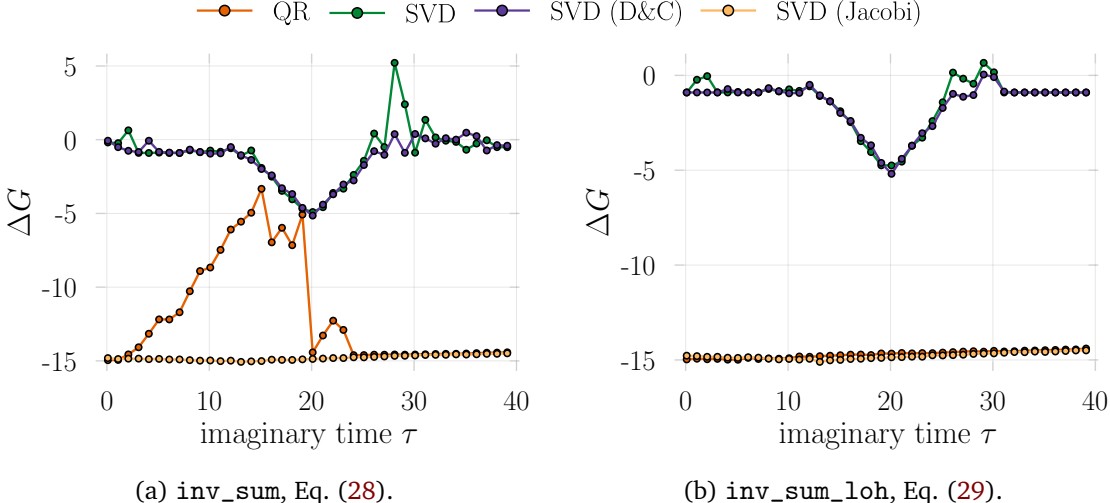

<table>
<tr><td>(a) <code>inv_sum</code>, Eq. (28).</td><td>(b) <code>inv_sum_loh</code>, Eq. (29).</td></tr>
</table>

Figure (6)  **Accuracy of the time-displaced Green's function** obtained from stabilized computations using the listed matrix decompositions and inversion schemes. Shown is $\Delta G = \log(\max(\mathrm{abs}(G(\tau,0) - G_{\mathrm{exact}}(\tau,0))))$ for $\beta = 40$.

## 6.2 Benchmarks

### 6.2.1 Accuracy

In Fig. 6, we show the logarithmic, maximal, absolute deviation of the time-displaced Green's function from the arbitrary precision result as a function of time-displacement $\tau$ at inverse temperature $\beta = 40$. Focusing on the inversion scheme <code>inv_sum</code> first, Fig. 6a, both regular and D&C SVD clearly fail to capture the intrinsic scales sufficiently and errors much beyond floating point precision are visible. Compared to these SVD variants, the QR decomposition systematically leads to equally or more accurate results. However, it clearly fails to be reliable at long times $\tau \sim \beta/2$ (the Green's function is anti-periodic in $\tau$). Only the Jacobi-method based SVD produces accurate Green's function values for all considered imaginary times.

Switching to the inversion scheme <code>inv_sum_loh</code>, the situation changes, as illustrated in Fig. 6b. While the non-Jacobi SVDs still have insufficient accuracy, the result for the QR decomposition improves dramatically compared to <code>inv_sum</code> and leads to stable Green's function estimates up to floating point precision along the entire imaginary time axis.

Similar to our findings for the equal-time Green's function, this suggests that only the Jacobi SVD and the QR decomposition are reliable for DQMC. The latter, however, must be paired with the <code>inv_sum_loh</code> inversion scheme to be reliable. To the best of our knowledge, this finding has not yet been mentioned in the literature.

### 6.2.2 Efficiency

Finally, we compare the computational runtime cost associated with both stable approaches: the Jacobi SVD combined with the regular inversion and the QR decomposition paired with <code>inv_sum_loh</code>. As shown in Fig. 7, we find that the latter is consistently faster for all considered system sizes. In relative terms, the SVD based approach falls behind by at least a factor of two and seems to display inferior scaling with the chain length $N$. This indicates that QR decompositions should be preferred over singular value decompositions when computing time-displaced Green's functions, in spite of the need to use an inversion scheme of higher complexity.

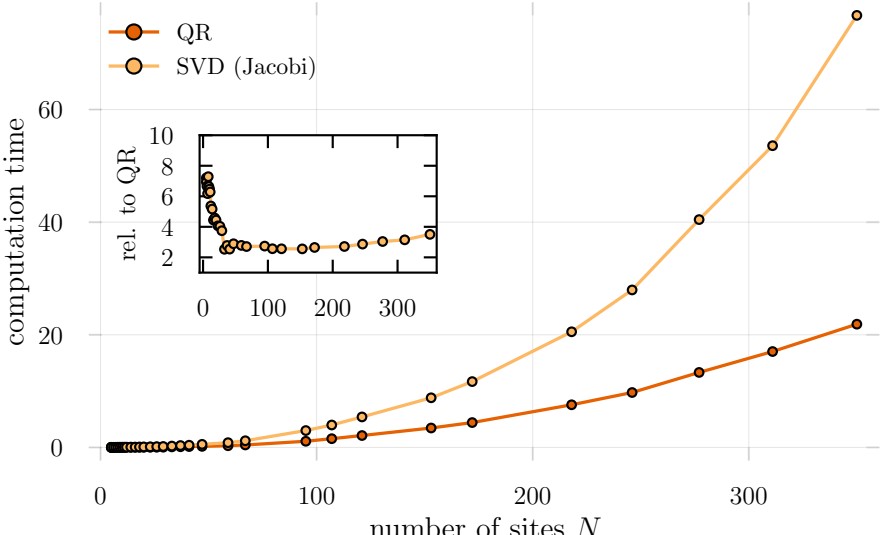

Figure (7) **Efficiency of the time-displaced Green's function** obtained from stabilized computations using the QR decomposition in combination with the inversion scheme `inv_sum_loh`, Eq. (29) and the Jacobi SVD paired up with the regular inversion scheme `inv_sum`, Eq. (28). Measurements are taken over multiple runs at $\tau = \beta/2 = 20$. The inset show the slowdown of the Jacobi SVD relative to the QR based approach.

## 7 Discussion

Numerical instabilities arise naturally in finite machine-precision quantum Monte Carlo simulations of many-fermion systems. Different schemes based on matrix factorizations have been proposed to handle the intrinsic exponential scales underlying these instabilities in a stable manner. As we have shown in this manuscript, these techniques can have vastly different accuracy and efficiency rendering them more or less suited for determinant quantum Monte Carlo simulations.

For our test system, the one-dimensional Hubbard model, we find that conventional and divide-and-conquer based singular value decompositions consistently fail to produce accurate equal time and time-displaced Green's functions, in particular at the lowest considered temperatures, $\beta \sim 40$. Only the QR decomposition and the Jacobi-method based SVD are able to stabilize the computation and produce reliable results. Importantly, we observe that in case of the time-displaced Green's function, the QR must be paired with an inversion scheme put forward by Loh *et al.* [18], an observation that, to the best of our knowledge, has not been mentioned in the literature before. No such qualitative dependence on the inversion procedure is observed for the Jacobi SVD.

In terms of efficiency, we find that the QR decomposition outperforms the Jacobi SVD by a large margin when utilized in stable Green's function computations. While expected from the fact that QR decompositions are computationally cheaper than SVDs, this difference is even apparent when the QR factorization is employed in a inversion scheme of higher complexity involving two (rather than one) matrix decompositions.

In summary, our empirical assessment demonstrates that, among the considered matrix factorizations and algorithms, the QR decomposition paired with the appropriate inversion schemes is the most efficient stabilization method for Green's function calculations in DQMC.

Finally, let us remark that the performance of any stabilization scheme is, in principle, model (parameter) dependent. While a systematic theoretical investigation is beyond the

scope of this manuscript, we include a brief analysis of a spin-fermion model for antiferromagnetic metallic quantum criticality in App. B. We therefore believe that our major conclusions bear some universality and can serve as a useful guide.

## Acknowledgements

We thank Peter Bröcker, Yoni Schattner, Snir Gazit, and Simon Trebst for useful discussions and Frederick Freyer for identifying a few typos in an early version of this manuscript. We acknowledge partial support from the Deutsche Forschungsgemeinschaft (DFG, German Research Foundation) Project No. 277101999–TRR 183 (project B01).

## A  Inversion schemes for time slice matrix stacks

In practical DQMC implementations one typically stores intermediate decomposed time slice matrix products $B(\tau_1, \tau_2)$ in a stack for reuse in future equal time Green's function calculations [7, 8, 37]. In this case, the inversion schemes in Eq. (17) needs to be prefixed by a stable procedure to combine two elements $U_L, D_L, X_L$ and $U_R, D_R, X_R$ from the stack, corresponding to $B_{l-1} \dots B_1$ and $B_M \dots B_l$ in Eq. 22. Below we describe the latter for both matrix decompositions considered in the main text[11].

### A.1  QR/UDT

(StableDQMC.jl: inv_one_plus(::UDT, ::UDT))

$$
\begin{aligned}
G &= \left[ \mathbb{1} + U_L D_L T_L \left( U_R D_R T_R \right)^\dagger \right]^{-1} \\
&= \left[ \mathbb{1} + U_L \underbrace{\left( D_L \left( T_L T_R^\dagger \right) D_R \right)}_{udt} U_R^\dagger \right]^{-1} \\
&= \left[ \mathbb{1} + UDT \right]^{-1},
\end{aligned}
\tag{30}
$$

with $U = U_L u$, $D = d$, and $T = t U_R^\dagger$. This $UDT$ factorization may then be substituted into Eq. (17).

### A.2  SVD

(StableDQMC.jl: inv_one_plus(::SVD, ::SVD))

$$
\begin{aligned}
G &= \left[ \mathbb{1} + U_L D_L V_L^\dagger U_R D_R V_R^\dagger \right]^{-1} \\
&= \left[ \mathbb{1} + U_L \underbrace{\left( D_L \left( V_L^\dagger U_R \right) D_R \right)}_{udv^\dagger} V_R^\dagger \right]^{-1} \\
&= \left[ \mathbb{1} + UDV^\dagger \right]^{-1},
\end{aligned}
\tag{31}
$$

with $U = U_L u$, $D = d$, and $V = v V_R$. This SVD factorization may then be substituted into Eq. (17).

---

[11]In StableDQMC.jl, Julia's multiple dispatch will automatically select the correct method based on the number of provided UDT factorizations.

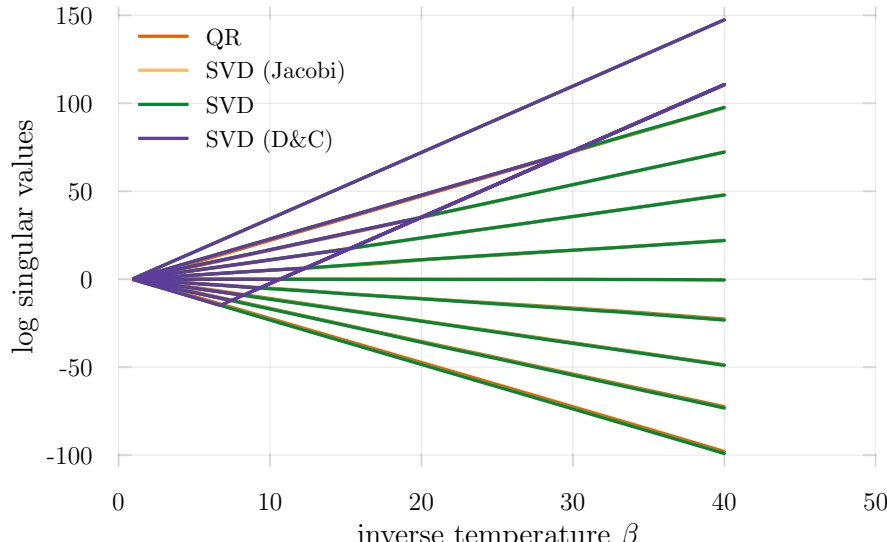

Figure (8)  **Comparison of matrix decompositions** in the calculation of the time slice matrix chain product $B_M B_{M-1} \cdots B_1$ near a metallic antiferromagnetic quantum critical point (the spin-fermion model considered in Ref. [1]). Different lines represent a selection[12]of logarithmic singular values as observed in the stabilized computations. The divide-and-conquer SVD (purple) shows large deviations at low temperatures $\beta \gtrsim 8$ ($\Delta\tau = 0.1$) while the other matrix decompositions are in approximate agreement (on top of each other).

## B  Spin-fermion model of a metallic quantum critical point

To assess the generality of the findings of the main text, we examine another model in which itinerant electrons are coupled to an antiferromagnetic order parameter. Specifically, we consider the two-dimensional square lattice spin-fermion model of Ref. [1] hosting a metallic quantum critical point (QCP), marking the onset of antiferromagnetic spin-density wave order at $T = 0$. As shown in extensive DQMC studies in Ref. [1], this system exhibits high-temperature superconductivity and features a non-Fermi liquid state in which fermionic quasiparticles lose their coherence.

We will focus our analysis on the vicinity of the QCP ($r = -1.74$) where quantum critical fluctuations are most pronounced and interactions are strong and relevant (in the renormalization group sense). The parameters are chosen as for the phase diagram in Fig. 2b of Ref. [1] with the exception of $L = 10$.

Analogous to Fig. 2 of the main text, Fig. 8 shows logarithmic singular values of the imaginary time slice matrix product chain $B_M \cdot B_{M-1} \cdots B_1$ stabilized by various matrix decompositions. Here, in contrast to Fig. 2, we drop the approximation $B_i = B$ and retain the full imaginary time dependence of the slice matrix factors. While the QR, Jacobi SVD, and regular SVD variants are capable of accurately capturing all intrinsic scales, the divide-and-conquer based SVD fails to reliably stabilize the matrix products and displays finite precision artifacts at low temperatures $T = 1/\beta \lesssim 0.125$.

Comparing these results to the findings of the main text — for the one-dimensional Hubbard model — we observe a qualitative deviation: The regular SVD appears to be more stable for the spin-fermion model. This is in spite of the fact that slice matrices near the metallic

---

[12]The four flavor (two spins, two bands) spin-fermion model with $L = 10$ leads to 400 singular values $s_i$. For simplicity, Fig. 2 shows only 10 of those, namely $s_1, s_{41}, s_{81}, ..., s_{361}$.

QCP are less well conditioned. A systematic investigation of the implementation details of LAPACK's regular SVD with respect to this disparity in accuracy for the two systems would be desirable, and is left for future work. Importantly, given that the SVD comes with a much higher computational cost, the major conclusion of the main text still holds for the strongly coupled spin-fermion model: Of all considered matrix factorizations, the QR decomposition is closest to the sweet spot of combined performance and accuracy.

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
