# Peer review of "Fast and stable determinant quantum Monte Carlo"

_SciPost Physics, doi:SciPost Phys. Core 2, 011 (2020)_

## Round 2 · Referee Report · Anonymous (Referee 1) · 2020-4-8

Strengths

1) Provides a systematic analysis of stabilization problems, and how to mitigate them, related to the determinant QMC method.

Weaknesses

1) It is not clear how general the results are.

Report

The paper considers the propagation of errors du to finite numerical precision in computing products of a large number of matrices, as appears in determinant QMC calculations. The stabilization problem and how to mitigate it has a long history, and here the author attempts to test several stabilization technique and compare them, both as to how well they work and as to their computational costs. The main conclusion is that the QR method is overall much better than the three SVD methods considered.
A potential limitation of the approach is that only a product of the same matrix is considered, which corresponds to a time-independent bosonic field in the determinant QMC simulation. In reality, the field of course fluctuates, and it is quite likely that the stabilization problems will be even worse for the typical contributing configurations, at least for some models (e.g., the Hubbard model with large U).
The paper is very technical, and it is not clear how general the results are. Nevertheless, the results and testing methodology may be interesting to researchers working on the technical aspects of the determinant QMC method. I recommend the paper to be published after revisions as listed below:

Requested changes

The illustration of the beta dependence of the singular values in Fig. 1 and some later figures requires some more explanation. I think the upper green line and lower orange line in Fig. 1 represents the smallest and largest singular values. But exactly what is the jagged data set that suddenly starts to increase with beta, taking off from the lower bound, e.g., around beta=10 in Fig. 1(a). Is that "curve" showing that the actual lowest singular value in the unstable numerical calculation? If so, it should be explained why there are no longer any small singular values. Perhaps it is somehow obvious that the destabilization causes the singular values to grow, but at least to me its not completely clear why.
As I mentioned above, the limitation to a product of the same matrix may give misleading results. Ideally, the author should at least show some results from an actual simulation, e.g., for the Hubbard model. otherwise it is not at all clear what the value is of this work. For instance, one could imagine that the QR method could work less well if the stabilization problem is even worse, as it may be for large U in typical Hubbard configurations.

  • validity: ok
  • significance: ok
  • originality: low
  • clarity: good
  • formatting: good
  • grammar: excellent

Author:  Carsten Bauer  on 2020-04-20  [id 800]

(in reply to Report 1 on 2020-04-08)

We thank the referee for his/her positive comments and his recommendation for publication. In the following, we reply to the questions raised by the referee.

Why are small singular values lost and why do they "grow" in Fig. 1? The fact that small singular values are lost and start to "grow" can be explained by the mixing with large ones: The finite machine precision limits the accuracy of a singular value relative to the largest singular value. When the difference surpasses floating point precision roundoff errors occur and the smallest singular values will try to be "as small as they can be" relative to the largest one. Note that the increase of the smallest singular values is precisely parallel to the growth of the largest one (both the lowest and topmost green lines have the same slope in Fig. 1) supporting this argument. A similar point holds for the loss of the smallest singular values, in Fig. 2: The LAPACK documentation, cited in the manuscript, explicitly mentions for the SVD error bounds: "Thus large singular values [...] are computed to high relative accuracy and small ones may not be." We will improve the caption text for the mentioned figures and will explain this point in more detail in a revised version of the manuscript. We thank the referee for pointing out this unclarity.

How general are the results? While this is an inherently difficult question, which asks for a systematic theoretical study beyond the scope of this manuscript, we agree that it would be beneficial to also present results for a more "real-world" model. To that end, we will add a new section to the appendix in which we consider a spin-fermion model for a metallic antiferromagnetic quantum critical point - concretely, the model studied in Ref. [1]. This appendix will contain the analogue of Fig. 1 for this strongly coupled system near quantum criticality, confirming our main finding that the QR decomposition is superior in terms of combined performance and accuracy.

We would again like to thank the referee for his/her comments and recommendation.

---

## Round 3 · Referee Report · Anonymous · 2020-5-16

Strengths

1) A careful study of the stability of the DQMC method, with comparisons of several stabilization methods.

2) The techniques are described in detail and are useful for those interested in the technical aspects of the DMRG method.

Weaknesses

1) The main paper shows detailed tests for a simplified case, but the behaviors found may not be very general (as indeed also shown in the new Appendix B).

Report

The author has made changes as recommended in my previous report. These changes have improved the paper considerably. It is particularly noteworthy that a new Appendix B has been added where an interesting spin-fermion model is considered as an example of a "real world" application of the kind I had suggested to test the generality of the results. The author finds that "The regular SVD appears to be more stable for the spin-fermion model", i.e., indeed the behaviors studied in the main paper do not appear to be completely general. Nevertheless, the paper is a valuable contribution and can now be published without further changes.

Requested changes

No further changes are required.

  • validity: good
  • significance: good
  • originality: ok
  • clarity: high
  • formatting: good
  • grammar: good

Author:  Carsten Bauer  on 2020-05-20

(in reply to Report 1 on 2020-05-16)

We would again like to thank the referee for his/her comments and his recommendation for publication.

---

## Round 3 · Referee Report · Anonymous · 2020-5-17

Report

This manuscript provides a detailed discussed of the issue of matrix product stabilisation, which is required in Determinantal QMC. Based on an open implementation in Julia the issue is illustrated and several solution strategies are carefully analysed both in terms of accuracy and time complexity. One central result within this study is that the QR decomposition is the "winner" among the considered matrix decompositions.

I like the open science aspect of this work and its pedagogical approach. I think the paper will be useful to newcomers in the field of Determinatal QMC.

Before publication I recommend the author to clarify the system size used in Fig. 1, or the clarify how the singular values were selected. Another minor issue concerns section 6.2.1., where starting on line 5 the sentence "Although the QR..." is unclear to me. I think the QR decomposition is compared here to the failing SVD implementations, but it is not clear from the formulation.

  • validity: good
  • significance: ok
  • originality: low
  • clarity: good
  • formatting: good
  • grammar: good

Author:  Carsten Bauer  on 2020-05-19

(in reply to Report 2 on 2020-05-17)

We thank the referee for his/her positive comments and his recommendation for publication. We will address the raised points by improving the manuscript in the follow way:

  • Clarifying the system size and number of (visible) singular values in the caption of Fig. 1.
  • Rephrasing the sentence in 6.2.1: As has been correctly pointed out by the referee, we are comparing the failing SVD variants and the QR approach in this line and this should be made unambiguously clear.

We would again like to thank the referee for his/her comments and recommendation.

---

## Round 3 · Referee Report · Anonymous · 2020-5-30

Strengths

1- A numerical study of the intricate parts of the auxiliary field quantum Monte carlo
algorithm that are often swept under the rug.

2- We like the fact that the author puts their code to the public
for reproducibility and submits his Journal to an Open-Access Journal.

Weaknesses

1- A weakness throughout the paper is that scans with respect to
the Hubbard interaction $U$ have been restricted to small/intermediate values.
This would also enhance the random fluctutions due to the bosonic fields as already
pointed out by another referee.

2- Method by Assaad used in ALF [ALF2017] for the measurement of time-displaced Green's functions was explicitely left out of the comparison.

Report

The paper is concerned with a
Numerical Assessment of various selected stabilization schemes
for evaluating equal-time and time-displaced Green's functions
in the realm of the auxiliary field QMC method
with respect to accuracy and speed.

The work is divided in three parts:
1.) First the problem of the stable calculation of forming a product of a long chain of matrices is considered.

Here the authors authors follow up on work done by [Bai 2011].
They compare methods using the SVD and a plain QR decomposition based technique and conclude that the QR decomposition is as accurate as the Jacobi SVD from lapack, a result already known from Bai (2011) where it was theoretically shown that both techniques are weakly backwards stable and the stability was also numerically observed.
The author also observes that non-jacobi methods from lapack fail to give
accurate results and traces this to the accuracy estimates given for lapacks SVD.
The deeper reason is that zgesvd (which implements the QR iteration)
and zgesdd are from the family of bidiagonalization algorithms where first a reduction to bidiagonal form using householder reflections is done and afterwards the respective methods get to work. This bidiagonalization step leads to a loss of relative accuracy for small singular
values. Jacobi's method does no bidiagonalization step and is able to recover small singular values with great accuracy [Demmel 1992]. This property very likely was also the reason that Bai et al used the Jacobi method for their numerical
assessment although they write in their main text "any stable method, e.g. the QR algorithms or the divide-and-conquer algorithm will be sufficient".
In that respect the author's paper clarifies that this has to be taken with caution.

The fact that zgesvd and zgesdd share the same bidiagonalization could also explain why their accuracy behaviour throughout the paper is very much alike and hence bounded by the bidiagonalization.

The author finishes with a plot comparing the relative speed w.r.t. to the QR decomposition and he observes a pronounced dependence on the system size $N$ of the Jacobi method.
This can be expected since the QR decomposition is entirely deterministic
in its runtime $\propto N^3$ whereas the Jacobi algorithm has a runtime $\propto S N^3$ where S is the number of jacobi sweeps.

We'd like to point out that lapack has a dgejsv routine that employs pivoting and a preconditioner to reduce the number of Jacobi sweeps and hence could significantly alter this picture.

2.) Accurate measurement of equal time Green's functions
Here a simple method and a method by Loh [Loh 2005, Loh 1989]
are compared. It would have been nice to include the method by
Bai [Bai 2011] which seems to get by without additional QR decompositions.
In terms of accuracy they observe similar behaviour which most likely
is due to the previously mentioned limitations of the SVD.

3.) Measurement of time-displaced Green's function.
Here they compare the accuracy obtained for measurements
of time-displaced Green's functions.
Equation (28) the author derives seem to have not been
published in the peer-reviewed literature but we note that it has been used in QUEST(http://quest.ucdavis.edu/)
at least during the last couple of years (https://github.com/Weisewill/quest-qmc/blob/master/SRC/dqmc_gtau.F90 around lines 550) together with a pivoted QR decomposition.

I consider it a weakness, that the method by Assaad used in ALF [ALF2017]
was explicitely left out of the comparison.

4.) Conclusion
The author closes the paper with the rather bold assessment
gathered from data on a one-dimensional Hubbard chain that
" the QR decomposition paired with the appropriate inversion schemes is the optimal stabilization method for Green’s function calculations in DQMC as it is both fast and stable."
While this sentiment is shared by the codes of ALF(http://alf.physik.uni-wuerzburg.de/)
and QUEST, I advise to a more cautionary wording here. While it is true that numerical evidence suggests that pivoted QR is stable enough for all our purposes the theoretical
assessment by Bai et al exposes the QR decomposition as the algorithm with lesser stability. The newly added appendix also hints at a slightly more cautionary wording.
With respect to speed we like to note that Jacobi algorithm exposes a lot of potential for blocking techniques and parallelization. I expect to see a lot of improvements coming there in the future.

These are my thoughts on the paper, I offer them to the author for consideration.

[Demmel 1992] https://doi.org/10.1137/0613074
[Bai 2011] https://doi.org/10.1016/j.laa.2010.06.023
[ALF 2017] https://scipost.org/SciPostPhys.3.2.013

Requested changes

Before Publication I recommend to the author the following changes:
1- chapter 4.2: Note that gesvd is the bidiagonal QR Iteration algorithm
2- The results using the SVD are dependent on the particular chosen algorithm. So I think it would be whorthwhile to at least cite a review of the current SVD algorithms and the active research performed in that field, e.g. Dongarra2018: https://doi.org/10.1137/17M1117732
3- Chapter 4.1 would benefit from a note that the idea of matrix factorizations dates back at least to the work of Loh.
4- section 4.2.2 would benefit from a note that pivoted QR is also the method of choice for two popular Codes: ALF [ALF2017] and QUEST(http://quest.ucdavis.edu/)

  • validity: good
  • significance: ok
  • originality: low
  • clarity: high
  • formatting: perfect
  • grammar: perfect

Author:  Carsten Bauer  on 2020-06-02

(in reply to Report 3 on 2020-05-30)

We thank the referee for his/her positive comments and his recommendation for publication.

We will address the raised points in a revised version of the manuscript. In particular, we will incorporate the requested changes and weaken our final conclusion in the discussion section according to the referee's comments.

---

## Round 3 · Author Response

We thank the referee for his/her positive comments and his recommendation for publication. Below we list all the changes made to the manuscript to take into account the points raised by the referee.

---

## Round 3 · List of Changes

* On page 5, the caption of Fig. 1 has been revised to clarify what is shown in the figure. In particular, the sentence "Different lines represent logarithmic singular values as observed in naive (green) and arbitrary precision computations (orange)." has been added.

* On page 8, the caption of Fig. 1 has been revised to clarify what is shown in the figure. Concretely, color references to the data in the figure and the sentence "Different lines represent logarithmic singular values as observed in stabilized computations." have been added.

* In the last paragraph on page 8, additional information on the relative error bounds, explaining the loss of small singular values in unstable computations, have been added: "The error is bounded relative to the largest singular value. Thus, large scales are computed to high relative accuracy and small ones may not be."

* On page 16, in the last paragraph of the "Discussion" section, we updated our statement about the model dependence of the findings to include a pointer to the new appendix on the spin-fermion model (see below).

* On page 17 an 18, we have added a new appendix "Spin-fermion model of a metallic quantum critical point". In this new appendix, we present a brief analysis of the stability of different matrix decompositions in the context of a more "real-world" example confirming the major conclusions of the main text.

---

## Round 6 · Author Response

We thank the referees for their positive comments and their recommendation for publication. Below we list all the changes made to the manuscript to take into account the points raised by the referees.

---

## Round 6 · List of Changes

* On page 5, we clarified the system size and number of (visible) singular values in the caption of Fig. 1.

* On page 6, in the first paragraph, we added a note that the matrix decomposition based stabilisation idea has already been raised by Loh et al. in 1989.

* On page 7, in the SVD section, we now cite Dongarra et al. (Ref. 32) as a reference for research on SVD algorithms.

* On page 7, in the SVD section, we indicate that the `gesvd` variant is a bidiagonal QR iteration scheme.

* On page 7, in the QR section, we added a note that the popular DQMC implementations ALF and QUEST (which we included as references) use the pivoted QR.

* On page 14, we rephrased two sentences in first paragraph of section 6.2.1 to clarify that we are comparing the QR to the failing SVD variants.

* On page 16, in the second to last paragraph of the discussion section, we weakened the previous statement about the QR being "optimal" and "fast and stable".

* On page 16, we added a note to the acknowledgments that this work has received partial support by the Deutsche Forschungsgemeinschaft (DFG, German Research Foundation).

---

## Editorial Decision

published